neuroscience

MRS, methodology, spectroscopy, MRI, data visualization, robust

**Author for correspondence:**
Niall W. Duncan
e-mail: niall.w.duncan@gmail.com

# Suggestions for improving the visualization of magnetic resonance spectroscopy voxels and spectra

## Vuong Truong[1,2] and Niall W. Duncan[1,2]

[1]Graduate Institute of Mind, Brain and Consciousness, Taipei Medical University, Taipei, Taiwan
[2]Brain and Consciousness Research Centre, TMU-ShuangHo Hospital, New Taipei City, Taiwan

 NWD, 0000-0002-3225-9119

Magnetic resonance spectroscopy (MRS) has seen an increase in popularity as a method for studying the human brain. This approach is dependent on voxel localization and spectral quality, knowledge of which are essential for judging the validity and robustness of any analysis. As such, visualization plays a central role in appropriately communicating MRS studies. The quality of data visualization has been shown to be poor in a number of biomedical fields and so we sought to appraise this in MRS papers. To do this, we conducted a survey of the psychiatric single-voxel MRS literature. This revealed a generally low standard, with a significant proportion of papers not providing the voxel location and spectral quality information required to judge their validity or replicate the experiment. Based on this, we then present a series of suggestions for a minimal standard for MRS data visualization. The primary point of these is that both voxel location and MRS spectra be presented from all participants. Participant group membership should be indicated where more than one is included in the experiment (e.g. patients and controls). A set of suggested figure layouts that fulfil these requirements are presented with sample code provided to produce these (github.com/nwd2918/MRS-voxel-plot).

## 1. Introduction

Single-voxel magnetic resonance spectroscopy (MRS) has seen an increase in popularity as a method for studying the human brain. The technique allows the *in vivo* estimation of metabolite concentrations within specific brain regions, opening up a variety of interesting research questions. For example, researchers have used MRS to draw connections between individual differences in psychological processes, such as working memory [1,2] or

inhibitory control [3,4], and neurotransmitter concentrations in a variety of brain regions [5]. This complements a longstanding use of MRS to study brain disorders. Here, metabolite levels have been compared between patients and controls, revealing neurochemical differences in conditions such as depression [6], schizophrenia [7], autism [8] and ADHD [9].

In both these types of study—looking at regional correlates of individual differences or comparing different groups—the data being analysed must necessarily come from the same part of the brain. If they do not then any correlations become difficult to interpret and spurious differences between groups may arise through systematic differences in the location sampled. Studies have, however, demonstrated test–retest voxel overlaps of only around 65% for PRESS voxels [10] and 75% for MEGA-PRESS [11]. This leaves scope for there to be relatively high location variability across participants and, by extension, for group differences in location to arise. Given this, it is important when publishing work for readers to be given information about the degree of overlap achieved in a study. It is not possible to judge the robustness of the work without this information, nor is it possible to identify any confounding factors arising from differences in voxel locations across participants.

A second key factor for interpreting MRS studies is the quality of the data included. Although the use of summary statistics about the spectra themselves (e.g. signal-to-noise ratio) and the quality of the analysis fit (e.g. fit error) is essential, they do not indicate the presence of certain artefacts and so final judgements must generally also involve visual inspection [12]. The information that such judgements rely upon should not, however, be available only to the researchers involved. As with voxel locations, it is important that the readers of published work also be able to directly judge the quality of the data included.

In both cases, the issue for readers is essentially one of data visualization. An effective representation of voxel locations would include information about all participants included in the analysis, presented in such a way as to show localization consistency. Where two or more groups are involved, an effective visualization would also present information about group-specific locations. For MRS spectra, an appropriate visualization would present the data of all participants. Group membership would be important additional information here too, to allow any patterns in data quality between groups to be identified. Providing all this information then allows the reader to appropriately interpret results and judge their reliability.

This need for effective data visualization to increase the interpretability and robustness of results has been highlighted in other areas of science [13]. This includes the brain imaging field, where there have, for example, been calls for more informative data visualization in EEG studies [14], along with various discussions about how to represent fMRI data in a robust and informative manner [15,16]. Unfortunately, the quality of data visualization has been found to be low in a significant proportion of publications across the biomedical literature [13,17,18].

## 2. Methods

### 2.1. Literature survey

Given the low quality of data visualization in other areas, to what extent do MRS publications currently provide the information described above as being necessary to appropriately interpret them? To gauge this, we conducted an informal appraisal of the recent single-voxel MRS literature and rated the data visualizations provided in 100 papers that compared individuals with mental or behavioural conditions to neurotypical groups and noted a range of reporting factors for each. Work that compared groups was chosen, as MRS studies in these areas are relatively common and are potentially affected by systemic location differences or spectral quality issues.

In more detail, we searched the Pubmed database with the following search term: '(MRS OR (magnetic resonance spectroscopy)) AND (depression OR depressive OR schizophrenia OR schizophrenic OR bipolar OR hyperactivity OR ADHD OR autism OR autistic)'. Only original human research papers written in English and published in the past 5 years (from 1 January 2015 to 31 December 2019) were considered. This search produced 596 results. These results were then randomly sorted and the first 100 that met the following criteria retained: (i) be an MRS study that used a single-voxel spectroscopy technique, and (ii) report an MRS result for at least one psychiatric disorder. A list of the included papers and details of their visualizations can be found in the electronic supplementary material.

Having selected a sample of 100 papers, we first appraised how well a reader could judge the consistency of voxel locations across participants and between groups. Visual representations of voxel locations, when present, were categorized according to whether the location was illustrated using real

data or whether an example location was drawn on manually. Ambiguous cases were marked as being from data (2/100 papers). Where real data was used, we further categorized the type of representation according to whether a single example participant was shown or whether data from the whole group was shown. The best case would be a figure created from all participant data that represented the degree of group overlap in some way.

To complement this information about visual representations, we also appraised textual descriptions of the methodology used to site the MRS voxels. These were graded into three categories: Category 1 reporting consisted of a non-specific location such as the lobe or general brain region targeted (e.g. occipital lobe). Category 2 reporting included a description of the specific region covered by the voxel (e.g. the voxel covered the pre- and post-central gyri). Finally, Category 3 reporting included the specific anatomical landmarks used for locating the voxel and how it related to these (e.g. 'the anterior edge was aligned with the genu of the corpus callosum'). How the voxel size was reported was also noted, along with whether any location coordinates or information about voxel overlap was given. Ideally, papers would provide a Category 3 methodological description, along with the voxel dimensions.

The second area that we appraised was how well a reader could judge the spectral quality for the data included in the study. For this, the visual representations of MRS spectra (where present) were categorized according to whether there was an example spectrum from a single participant; a figure showing the average spectra of a group or groups; or a figure presenting the spectra from all participants. The latter visualization would be the optimal way of representing the data.

## 2.2. Data for figures

To illustrate approaches to visualization that increase the amount of information given to the reader, we created a number of example figures. The spectral data for these were taken from the publicly available 'Big GABA' dataset [19]. MEGA-PRESS data from two sites (GE-4 and GE-7) were used, with sites treated as groups in figures where group comparisons are illustrated. Individual MRS voxels are not available with this dataset and so simulated voxels covering the posterior cingulate cortex (PCC)/precuneus region were created by situating a set of $30 \times 30 \times 30 \, \text{mm}^3$ binary masks on the MNI152 template.

# 3. Results

## 3.1. Voxel location

The majority of studies provided a figure representing the voxel location but 12% did not. For those that did provide a figure, the most common approach was to create an illustration of the location by drawing over an anatomical image (46%). This was closely followed by showing the true location for one participant (i.e. a voxel mask created from the data) superimposed upon their anatomical image (40%). In terms of representing the variability of voxel locations between participants, very few studies showed the location of more than one person: one study showed the location for each participant and one study provided a representation of the group overlap.

In terms of the methods text, the largest proportion of studies provided a Category 1 description of the voxel location (56%). In other words, their methods only related a non-specific anatomical location with no details of landmarks used during scanning. The second most common type of description was then Category 2 (28%), giving a description of the specific brain region that the voxel covered. The remaining studies gave a Category 3 description (16%), meaning a specific location was provided, often along with a description of the anatomical landmarks used to locate the voxel during scanning. In a few cases (2/100), the centre-of-mass coordinates in MNI space for all voxels were reported. When reporting the size of the voxel, the most common approach was to report the dimensions of each side (89%). The remaining studies reported the voxel volume (6%) or, in some cases, gave no information at all (5%). No studies reported the degree to which the MRS voxels for different participants overlapped with each other, nor did any studies that compared different groups report how similar the voxel locations for each group were to each other.

## 3.2. Visual representation of spectra

Around a fifth of studies did not provide any visual representation of the MRS spectra (19%). Of those that did, the majority showed an example spectrum from a single participant (77%), along with a small

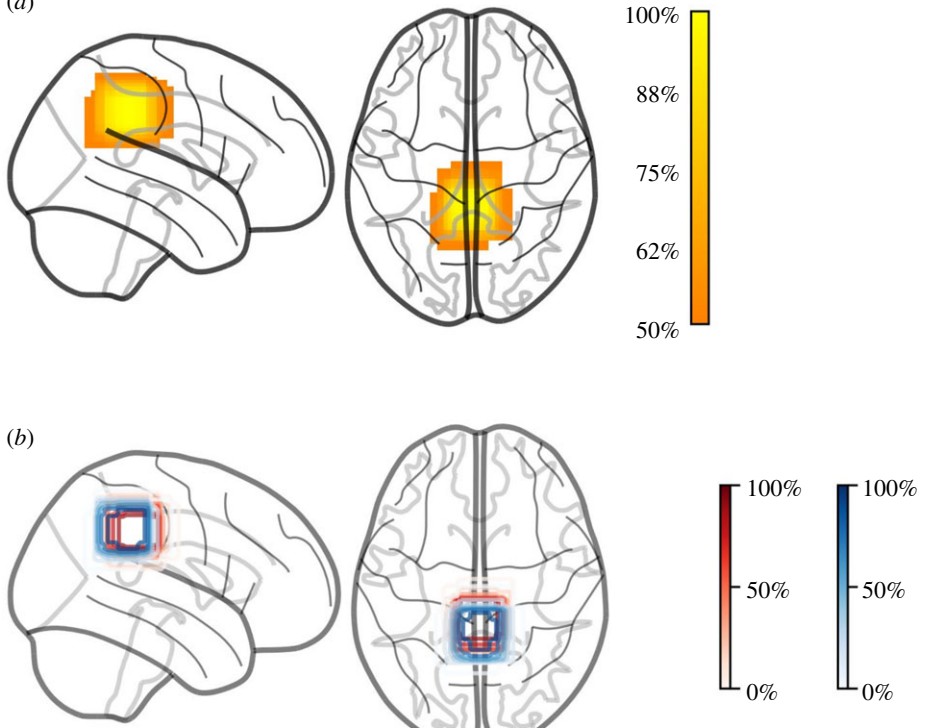

**Figure 1.** Voxel overlap. Individual MRS voxels are converted to MNI space and then combined to show the overlap between participants. (*a*) A voxel overlap density map for a single group of participants. The heatmap shows the percentage overlap of participant MRS voxels at each point in the brain. (*b*) Voxel overlap densities for two separate groups, as denoted by red and blue contours. The shading of the contour lines represents the percentage overlap at that point. This figure type can be extended to include multiple participant groups. Here, we can see that the two groups have somewhat different patterns of voxel placement. Brain outlines are created using the nilearn toolbox [26].

number that showed a group average (3%). Only two studies provided a visual representation of the spectra for all participants. These two papers were also the only ones to provide figures that allowed a comparison of the spectra between the groups as a whole. Notably, the studies that presented the data from a single participant did not provide any rationale for why that participant was selected.

# 4. Discussion

From this appraisal of the human single-voxel MRS literature we can see that there is a generally poor standard of reporting of voxel locations and spectral quality. The low standard in many papers makes proper interpretation of the results or replication of the studies difficult or, in some cases, impossible. To improve this situation, we suggest a minimal standard for the visualization of MRS voxels and spectra. This is intended to complement other work discussing optimal approaches to MRS data acquisition and analysis [20,21], as well as a recently proposed approach to assessing the overall quality of reported MRS studies [22].

## 4.1. Voxel location

Images representing voxel locations should be produced from individual MRS voxel masks obtained from the data of all participants and transformed into standard space. Such masks can be created automatically from most data types using a number of software tools [10,23] and are already required for partial volume correction, a necessary part of most analysis pipelines [24,25]. Alignment to a standard space (e.g. MNI152) can be done via individual anatomical images with any MRI analysis package, including FSL, SPM and AFNI. The masks in standard space can then be combined to create a voxel overlap density map (figure 1). This shows at each standard space anatomical voxel the percentage of overlap between the set of participant masks. With this, one can judge the consistency

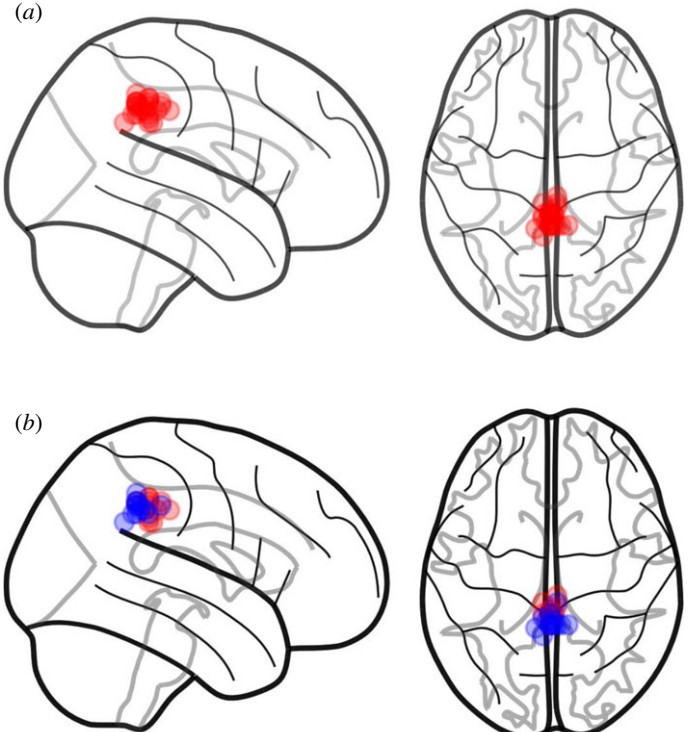

**Figure 2.** Voxel centroids. MRS masks in MNI space are used to calculate the voxel centroid for each participant. (*a*) The centroid for a single group of participants. It is easy to see how tightly clustered MRS voxels are around a target anatomical location. (*b*) The centroids for two separate groups, as denoted by red and blue colours. Here, we can see evidence for a systematic difference in clustering between the two groups. Brain outlines are created using the nilearn toolbox [26].

of voxel placement in the study and gauge the specific anatomical regions from which MRS data were recorded. Such maps have been used in a number of previous studies [11,27,28]. In the case of group comparison studies, separate density maps can be created and presented to help identify where there may be a systematic difference in location between the groups. It should be noted, however, that standard space alignments can be problematic in groups with extensive cortical atrophy, such as Alzheimer's disease or stroke patients, and so care should be taken if applying this approach in such contexts.

Complementary information about voxel locations can be presented through plots of mask centre-of-mass. Here, the centre-of-mass of each participant's standard space mask is calculated and then displayed on a standard space brain image (figure 2). Calculations can be performed in a range of tools, including, for example, FSL's fslstats, Python or Matlab. This approach is particularly useful in group comparison studies as it more clearly shows any pattern in the locations of voxels. With this, any systematic differences between groups can be easily identified. Statistical analyses can also be applied to the location distributions, although this is beyond the scope of the current work.

## 4.2. Spectral quality

The majority of MRS analysis tools (e.g. LCModel, TARQUIN, Gannet) have options to output text files containing the processed MRS spectra. These can then be used to create figures that combine the spectra from all participants in any plotting software. Although a group average spectrum with error bars would be an acceptable way to present this data, also showing individual spectra is optimal as this allows the reader to fully judge data quality. An example of such a plot is shown in figure 3. This plot also demonstrates how data from multiple groups can be shown on a single figure so that data quality can be compared between them. This is a key step, particularly in comparisons of patients and controls where there may be systematic differences in factors such as head motion that can affect MRS data quality [29]. Finally, it would be optimal to provide images of any spectra that are rejected as supplementary figures or hosted online (e.g. at https://figshare.com/) as this makes the spectra selection process transparent to the reader.

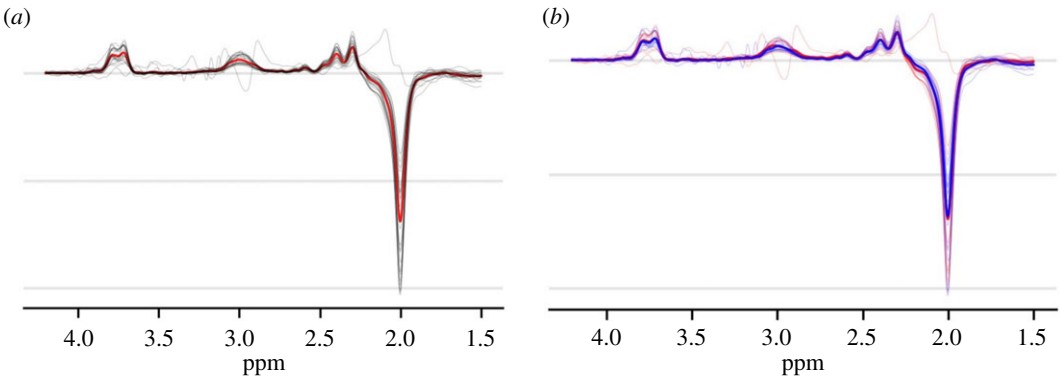

**Figure 3.** MRS spectra. Individual MRS spectra are shown along with the average spectra for a particular group. (*a*) MEGA-PRESS difference spectra for a single group. Individual spectra are shown in black with the group average added in red. (*b*) The same for two separate groups, as denoted by red and blue colours. Here, individual spectra are shown in the relevant group colour with the group average added as a thicker line of the same colour. Note that showing the individual data reveals one participant with invalid data.

## 4.3. Figure code

Code to create the suggested figures is available at https://github.com/nwd2918/MRS-voxel-plot and https://mrshub.org/software_visualization. This is written in Python and is fully open-source, making it accessible to all researchers. The code is designed to be simple to use, such that a researcher need only add the relevant directory and file names, with all other steps being automatic. Usage instructions are provided with the code.

## 5. Conclusion

Interpretation and replication of studies requires that sufficient information be communicated to the scientific audience. An informal appraisal of the human single-voxel MRS literature revealed generally low-quality representations of essential information, specifically voxel locations and spectral quality. To help improve this, we set out some basic visualization approaches that give readers greater information about the data included in a study to better judge its robustness and validity (in conjunction with quality measures such as SNR and FWHM). Particular focus was made on studies that compare patients and controls, where there may be systematic differences in localization and quality between groups. We hope that these suggestions and the code provided will be a useful resource that contributes to an improvement in the reporting of MRS studies.

Data accessibility. Literature appraisal data are provided as supplementary material. Data and relevant code for this research work are stored in GitHub: https://github.com/nwd2918/MRS-voxel-plot and have been archived within the Zenodo repository: https://doi.org/10.5281/zenodo.3904278.
Authors' contributions. N.W.D. conceived of the study; V.T. collected and analysed data; N.W.D. and V.T. wrote the code; N.W.D. and V.T. prepared the manuscript; all authors gave final approval for publication and agree to be held accountable for the work performed therein.
Competing interests. The authors declare no conflicts of interest.
Funding. This work was supported by funding from the Taiwan Ministry of Science and Technology to N.W.D. (107-2410-H-038-004-MY2; 108-2410-H-038-008-MY2).
Acknowledgements. The authors are grateful to Elizabeth McManus (University of Manchester) for useful comments on a draft of this paper.

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
