## [Reviewer comments · Royal Society Open Science]

Review History

RSOS-200600.R0 (Original submission)

Review form: Reviewer 1

Is the manuscript scientifically sound in its present form?

No

Are the interpretations and conclusions justified by the results?

No

Is the language acceptable?

Yes

Do you have any ethical concerns with this paper?

No

Have you any concerns about statistical analyses in this paper?

No

Recommendation?

Reject

Comments to the Author(s)

The manuscript is about how to visualize VOI placement and display data in MRS acquisitions. In my opinion, the most important thing in MRS is spectral data quality. As such, I do not see the importance of this manuscript in this field.

VOI placement are usually done by technologist or researcher. These personnel are trained by their supervisor using landmarks in order to place a VOI at the right location. Based on the 100 literatures and what was or was not reported, the authors are assuming the VOI was not correctly placed. Same VOI location does not necessarily mean same VOI content due to variation in brain shape between individual. As such, I do not see the importance of showing the mean VOI location from all subjects. Using an insert to show the VOI placement on an MRI image is deemed sufficient in my opinion for VOI visualization. Showing a centroid (Fig 2) is more confusing and hard to interpret due to overlap and as such much more difficult to see VOI overlap.

The most important result in MRS is the the data quality and reproducible of the data. If the spectral quality is not good or have contamination, this will result in unusable or wrong metabolite concentrations. I believe showing the mean+-std deviation of all spectra is the recommendation from the MRS consensus group, which will soon be published. A such, I do not see the point of this manuscript.

Review form: Reviewer 2

Is the manuscript scientifically sound in its present form?

Yes

Are the interpretations and conclusions justified by the results?

Yes

Is the language acceptable?

Yes

Do you have any ethical concerns with this paper?

No

Have you any concerns about statistical analyses in this paper?

No

Recommendation?

Accept as is

Comments to the Author(s)

The manuscript describes a longstanding problem within the magnetic resonance spectroscopy literature. It peels back the curtain to reveal there are major problems with how data is acquired and incorporated into published analyses. The authors design a review of the recent literature in a key field publishing magnetic resonance spectroscopy studies detailing the scope of the problems using defined criteria. Subsequently, they offer solutions from the larger imaging field to address the problems.

The article was very easy to read and understand. The figures were excellent.

Review form: Reviewer 3 (Paul Mullins)

Is the manuscript scientifically sound in its present form?

Yes

Are the interpretations and conclusions justified by the results?

Yes

Is the language acceptable?

Yes

Do you have any ethical concerns with this paper?

Yes

Have you any concerns about statistical analyses in this paper?

No

Recommendation?

Accept with minor revision (please list in comments)

Comments to the Author(s)

I would first like to apologise to the Authors for the delay in reading and reviewing this paper.

I have read the paper and agree with its premise 100%. There is currently a minimum reporting guidelines paper for MRS coming out from the MRS community, and this paper would be an excellent addition to the reference list for that Report, and complements it well. (Goes beyond it in presenting a good way to present MRS data.)

There is an omission in the introduction though that I think the authors should address. This manuscript is dealing only with human brain single voxel spectroscopy (SVS). This should be made clear at the outset, but this should be an easy enough fix. I don't know if the authors have considered modifying the code to work with animal SVS, SVS from other body regions, or MRSI data, but it is something they might consider for future projects.

I have a few comments and concerns over the origin of the data used to generate the figures. The figures themselves are very nice, and appropriate, but I think the authors should describe if the images represent real or simulated data, and if real which data set they come from.

I see that the software used to generate the figures is also available on <https://mrshub.org/> - the authors should add that location into the manuscript, as it may also help other researchers find that resource.

While the software is accessible, the documentation is not as extensive as it could be. I'm thinking specifically of the novice MRS researcher (say a postgrad, or junior postdoc) who is tasked with getting this visualisation up and running by a principal investigator in their lab or institute. There is not quite enough detail to help this type of researcher get going quickly with the software. An eye to the novice in the documentation would be appreciated. I do not think this is enough to prevent publication, but it is something the authors may wish to think about if they want to make their software more widely used. An example data set, arranged as required for the software to work, with some guidance on data structure, could be hosted with the code, and might help in this regard.

No mention is made of how the data used to generate the figures was acquired, what study this data comes from, nor if it was acquired in accordance with the principles of the declaration of Helsinki. I expect it all was, but without a section detailing where and what this data actually is,

it would only be an assumption. My concern could be addressed by the authors providing this information in the manuscript - best place would be in the methods section.

Apart from these concerns and suggestions I found this paper very useful, and would like to start using the code in presentation of data from my own lab. I applaud the authors for the work, and hope my suggested corrections are not too onerous, so a resubmission can occur quickly.

Decision letter (RSOS-200600.R0)

Dear Dr Duncan,

The editors assigned to your paper ("Suggestions for improving the visualization of magnetic resonance spectroscopy voxels and spectra") have now received comments from reviewers. We would like you to revise your paper in accordance with the referee and Associate Editor suggestions which can be found below (not including confidential reports to the Editor). Please note this decision does not guarantee eventual acceptance.

Please submit a copy of your revised paper before 28-Jun-2020. Please note that the revision deadline will expire at 00.00am on this date. If we do not hear from you within this time then it will be assumed that the paper has been withdrawn. In exceptional circumstances, extensions may be possible if agreed with the Editorial Office in advance. We do not allow multiple rounds of revision so we urge you to make every effort to fully address all of the comments at this stage. If deemed necessary by the Editors, your manuscript will be sent back to one or more of the original reviewers for assessment. If the original reviewers are not available, we may invite new reviewers.

- Data accessibility

<http://datadryad.org/submit?journalID=RSOS&manu=RSOS-200600>

- Competing interests

- Authors' contributions

- Acknowledgements

- Funding statement

Kind regards,

Andrew Dunn

on behalf of Dr Rochelle Ackerley (Associate Editor)

Associate Editor's comments (Dr Rochelle Ackerley):

Associate Editor: 1

Comments to the Author:

Three expert reviewers have provided comments on your manuscript. Although two were positive, one reviewer did have major concerns about the reliability and quality of the data presented. This needs to be accounted for and may be a limitation of the work. Reviewer 3 also had concerns about the origin of the data, which need to be addressed. Further, more detail is needed about the software, as for it to be useful to the imaging community, it must be accessible and easy to implement.

Comments to Author:

Reviewers' Comments to Author:

Reviewer: 1

Comments to the Author(s)

The manuscript is about how to visualize VOI placement and display data in MRS acquisitions. In my opinion, the most important thing in MRS is spectral data quality. As such, I do not see the importance of this manuscript in this field.

VOI placement are usually done by technologist or researcher. These personnel are trained by their supervisor using landmarks in order to place a VOI at the right location. Based on the 100 literatures and what was or was not reported, the authors are assuming the VOI was not correctly placed. Same VOI location does not necessarily mean same VOI content due to variation in brain shape between individual. As such, I do not see the importance of showing the mean VOI location from all subjects. Using an insert to show the VOI placement on an MRI image is deemed sufficient in my opinion for VOI visualization. Showing a centroid (Fig 2) is more confusing and hard to interpret due to overlap and as such much more difficult to see VOI overlap.

The most important result in MRS is the the data quality and reproducible of the data. If the spectral quality is not good or have contamination, this will result in unusable or wrong metabolite concentrations. I believe showing the mean+std deviation of all spectra is the recommendation from the MRS consensus group, which will soon be published. A such, I do not see the point of this manuscript.

Reviewer: 2

Comments to the Author(s)

The manuscript describes a longstanding problem within the magnetic resonance spectroscopy literature. It peels back the curtain to reveal there are major problems with how data is acquired and incorporated into published analyses. The authors design a review of the recent literature in a key field publishing magnetic resonance spectroscopy studies detailing the scope of the problems using defined criteria. Subsequently, they offer solutions from the larger imaging field to address the problems.

The article was very easy to read and understand. The figures were excellent.

Reviewer: 3

Comments to the Author(s)

I would first like to apologise to the Auhtors for the delay in reading and reveiwing this paper.

I have read the paper and agree with it's premise 100%. There is currently a minimum reporting guidelines paper for MRS coming out from the MRS community, and this paper would be an excellent addition to the reference list for that Report, and complements it well. (Goes beyond it in presenting a good way to present MRS data.)

There is an omission in the introduction though that I think the authors should address. This manuscript is dealing only with human brain single voxel spectroscopy (SVS). This should be made clear at the outset, but this should be an easy enough fix. I don't know if the authors have considered modifying the code to work with animal SVS, SVS from other body regions, or MRSI data, but it is something they might consider for future projects.

I have a few comments and concerns over the origin of the data used to generate the figures. The figures themselves are very nice, and appropriate, but I think the authors should describe if the images represent real or simulated data, and if real which data set they come from.

I see that the software used to generate the figures is also available on <https://mrshub.org/> - the authors should add that location into the manuscript, as it may also help other researchers find that resource.

While the software is accessible, the documentation is not as extensive as it could be. I'm thinking specifically of the novice MRS researcher (say a postgrad, or junior postdoc) who is tasked with getting this visualisation up and running by a principal investigator in their lab or institute. There is not quite enough detail to help this type of researcher get going quickly with the software. An eye to the novice in the documentation would be appreciated. I do not think this is enough to prevent publication, but it is something the authors may wish to think about if they want to make their software more widely used. An example data set, arranged as required for the software to work, with some guidance on data structure, could be hosted with the code, and might help in this regard.

No mention is made of how the data used to generate the figures was acquired, what study this data comes from, nor if it was acquired in accordance with the principles of the declaration of Helsinki. I expect it all was, but without a section detailing where and what this data actually is, it would only be an assumption. My concern could be addressed by the authors providing this information in the manuscript - best place would be in the methods section.

Apart from these concerns and suggestions I found this paper very useful, and would like to start using the code in presentation of data from my own lab. I applaud the authors for the work, and hope my suggested corrections are not too onerous, so a resubmission can occur quickly.

Author's Response to Decision Letter for (RSOS-200600.R0)

See Appendix A.

RSOS-200600.R1 (Revision)

Review form: Reviewer 1

Is the manuscript scientifically sound in its present form?

Yes

Are the interpretations and conclusions justified by the results?

Yes

Is the language acceptable?

Yes

Do you have any ethical concerns with this paper?

No

Have you any concerns about statistical analyses in this paper?

No

Recommendation?

Accept with minor revision (please list in comments)

Comments to the Author(s)

I would suggest the authors to add my points of concern in the text.

E.g. importance of spectral data quality and reproductivity in addition to having visualization

Review form: Reviewer 3 (Paul Mullins)

Is the manuscript scientifically sound in its present form?

Yes

Are the interpretations and conclusions justified by the results?

Yes

Is the language acceptable?

Yes

Do you have any ethical concerns with this paper?

No

Have you any concerns about statistical analyses in this paper?

No

Recommendation?

Accept as is

Comments to the Author(s)

I thank the authors for addressing my concerns, and wish them the very best with their future research

Decision letter (RSOS-200600.R1)

Dear Dr Duncan,

It is a pleasure to accept your manuscript entitled "Suggestions for improving the visualization of magnetic resonance spectroscopy voxels and spectra" in its current form for publication in Royal

Society Open Science. The comments of the reviewer(s) who reviewed your manuscript are included at the foot of this letter.

on behalf of Dr Rochelle Ackerley (Associate Editor)

Reviewer comments to Author:
Reviewer: 3

Comments to the Author(s)
I thank the authors for addressing my concerns, and wish them the very best with their future research

Reviewer: 1

Comments to the Author(s)
I would suggest the authors to add my points of concern in the text.
E.g. importance of spectral data quality and reproductivity in addition to having visualization ...

Appendix A

We would like to thank all the reviewers and the editor for their time and for their comments on our work. Our responses to these comments are given below in blue text. Changes are also marked in the manuscript in the same colour.

Associate Editor's comments (Dr Rochelle Ackerley):

Associate Editor: 1

Comments to the Author:

Three expert reviewers have provided comments on your manuscript. Although two were positive, one reviewer did have major concerns about the reliability and quality of the data presented. This needs to be accounted for and may be a limitation of the work. Reviewer 3 also had concerns about the origin of the data, which need to be addressed. Further, more detail is needed about the software, as for it to be useful to the imaging community, it must be accessible and easy to implement.

In brief, we have clarified in the manuscript which data were used to produce the example images. These data are now also available online along with the code so that people can work through the process of making the figures. In addition, we have added an online tutorial that goes through each step and provides additional detail to help others implement the approach (see <https://github.com/nwd2918/MRS-voxel-plot/tree/master/how-to>). We agree that this is a valuable addition and helps make the software more accessible. Full details of these changes are given below at the relevant points.

Comments to Author:

Reviewers' Comments to Author:

Reviewer: 1

Comments to the Author(s)

The manuscript is about how to visualize VOI placement and display data in MRS acquisitions. In my opinion, the most important thing in MRS is spectral data quality. As such, I do not see the importance of this manuscript in this field.

VOI placement are usually done by technologist or researcher. These personnel are trained by their supervisor using landmarks in order to place a VOI at the right location. Based on the 100 literatures and what was or was not reported, the authors are assuming the VOI was not correctly placed. Same VOI location does not necessarily mean same VOI content due to variation in brain shape between individual. As such, I do not see the importance of showing the mean VOI location from all subjects. Using an insert to show the VOI placement on an MRI image is deemed sufficient in my opinion for VOI visualization. Showing a centroid (Fig 2) is more confusing and hard to interpret due to overlap and as such much more difficult to see VOI overlap.

The most important result in MRS is the the data quality and reproducible of the data. If the spectral quality is not good or have contamination, this will result in unusable or wrong metabolite concentrations. I believe showing the mean+-std deviation of all spectra is the

recommendation from the MRS consensus group, which will soon be published. As such, I do not see the point of this manuscript.

We agree entirely with the reviewer that “the most important result in MRS is the data quality and reproducibility of the data”. Ensuring this was the motivation behind the work and the aim of the visualisation approach described.

The assumption behind the work is not that the scientists or clinicians acquiring MRS data would be incompetent in their placement of voxels. Instead we follow the fundamental tenet of the scientific method that one should not base one's assessment of results upon the authority of those presenting them but that one should form an opinion independently. To form such an opinion one needs information; information which our literature appraisal shows is not generally provided at present.

The reviewer is correct to point out that precise voxel contents differ between individuals. The logic of group studies requires, however, that anatomical locations have sufficient homogeneity that a comparison across individuals makes sense. Corrections based on individual differences in anatomy may be required in such comparisons but this is not the issue that we deal with in the current work. Problems arise, however, if anatomical locations are not matched across participants. Showing how well the locations match thus allows the reader to appraise how well the data fit with the background assumption that the same part of the brain is being compared in all participants.

We present different options for presenting voxel locations as one or the other may be better depending on the particular study. In the case of the centroid image the overlap of points is desirable as this would indicate consistent voxel placement. In our example image we present a case where two groups do not have entirely overlapping centroid clouds, indicating that there is a systematic difference in voxel location between them.

Showing the mean and standard deviation has been standard for some time in the scientific literature but, as discussed in the introduction, there has been a recent move towards displaying individual data also as this reveals more information. We present an example of that in our example figure where one participant has invalid data, something that would not be apparent in a plot of just the mean and standard deviation.

Reviewer: 2

Comments to the Author(s)

The manuscript describes a longstanding problem within the magnetic resonance spectroscopy literature. It peels back the curtain to reveal there are major problems with how data is acquired and incorporated into published analyses. The authors design a review of the recent literature in a key field publishing magnetic resonance spectroscopy studies detailing the scope of the problems using defined criteria. Subsequently, they offer solutions from the larger imaging field to address the problems.

The article was very easy to read and understand. The figures were excellent.

We thank the reviewer for their comments and are glad to receive such positive feedback.

Reviewer: 3

We thank the reviewer for their constructive comments and quite understand any delay given the current global situation. Our responses to individual comments are interspersed below.

Comments to the Author(s)

I would first like to apologise to the Authors for the delay in reading and reviewing this paper.

I have read the paper and agree with its premise 100%. There is currently a minimum reporting guidelines paper for MRS coming out from the MRS community, and this paper would be an excellent addition to the reference list for that Report, and complements it well. (Goes beyond it in presenting a good way to present MRS data.)

There is an omission in the introduction though that I think the authors should address. This manuscript is dealing only with human brain single voxel spectroscopy (SVS). This should be made clear at the outset, but this should be an easy enough fix. I don't know if the authors have considered modifying the code to work with animal SVS, SVS from other body regions, or MRSI data, but it is something they might consider for future projects.

This is a good point - we have now made it clearer in the abstract and main text that we only deal with human SVS data.

Expanding the features of the code so that it can work with different species and data types is certainly something that we would be interested in doing in the future. We do not yet have direct experience with data beyond human SVS, however, and so wouldn't be comfortable doing this at this point. The code is open and so there is the potential for others to contribute, something we hope will happen over time.

I have a few comments and concerns over the origin of the data used to generate the figures. The figures themselves are very nice, and appropriate, but I think the authors should describe if the images represent real or simulated data, and if real which data set they come from.

I see that the software used to generate the figures is also available on <https://mrshub.org/> - the authors should add that location into the manuscript, as it may also help other researchers find that resource.

This detail has been added to the manuscript:

“Code to create the suggested figures is available at <https://github.com/nwd2918/MRS-voxel-plot> and https://mrshub.org/software_visualization.”

While the software is accessible, the documentation is not as extensive as it could be. I'm thinking specifically of the novice MRS researcher (say a postgrad, or junior postdoc) who is tasked with getting this visualisation up and running by a principal investigator in their lab or institute. There is not quite enough detail to help this type of researcher get going quickly with the software. An eye to the novice in the documentation would be appreciated. I do not think this is enough to prevent publication, but it is something the authors may wish to think about if they want to make their software more widely used. An example data set, arranged as required for the software to work, with some guidance on data structure, could be hosted with the code, and might help in this regard.

This is a good suggestion - we admit it can be easy to overestimate the usability of code that one already knows well.

We have created walk-throughs on the github page for single group and multiple group analyses (<https://github.com/nwd2918/MRS-voxel-plot/tree/master/how-to>). These are in the form of jupyter notebooks and include example data so that the user can work through the process themselves.

No mention is made of how the data used to generate the figures was acquired, what study this data comes from, nor if it was acquired in accordance with the principles of the declaration of Helsinki. I expect it all was, but without a section detailing where and what this data actually is, it would only be an assumption. My concern could be addressed by the authors providing this information in the manuscript - best place would be in the methods section.

The data that we originally used to create the figures was from a study that we conducted recently. These data cannot be shared openly, however, as consent was not obtained from participants to do so. To get around this problem we have changed the figures so that they are based on data that can be shared.

For the spectra figures we now use data downloaded from the “Big GABA” project (<https://www.nitrc.org/projects/biggaba/>). Unfortunately, voxel masks are not currently available for these and so we created simulated masks located in the PCC/precuneus region these spectra are from. These masks were created manually on the MNI152 template. We have added these details to the manuscript as follows:

“Data for figures

To illustrate approaches to visualisation that increase the amount of information given to the reader, we create a number of example figures. The spectral data for these were taken from the publicly available “Big GABA” dataset (Mikkelsen et al., 2017). MEGA-PRESS data from two sites (GE-4 and GE-7) were used, with sites treated as groups in figures where

group comparisons are illustrated. Individual MRS voxels are not available with this dataset and so simulated voxels covering the PCC/precuneus region were created by situating a set of $30 \times 30 \times 30 \text{ mm}^3$ binary masks on the MNI152 template.”

By chance, the new data used for making the figures includes one spectrum that illustrates how poor quality data can appear in a group without it influencing the mean. We have highlighted this in the captions to Figure 3 as follows:

“Individual MRS spectra are shown along with the average spectra for a particular group. Figure (A) shows MEGA-PRESS difference spectra for a single group. Individual spectra are shown in black with the group average added in red. Figure (B) shows the same for two separate groups, as denoted by red and blue colours. Here, individual spectra are shown in the relevant group colour with the group average added as a thicker line of the same colour. Note that showing the individual data reveals one participant with invalid data.”

Apart from these concerns and suggestions I found this paper very useful, and would like to start using the code in presentation of data from my own lab. I applaud the authors for the work, and hope my suggested corrections are not too onerous, so a resubmission can occur quickly.